

# A simple grid implementation with Berkeley Open Infrastructure for Network Computing using BLAST as a model

Watthanai Pinthong[1], Panya Muangruen[2], Prapat Suriyaphol[3] and Dumrong Mairiang[4,5]

[1] Department of Anatomy, Faculty of Medicine Siriraj Hospital, Mahidol University, Bangkok, Thailand
[2] Siriraj Information Technology Department, Faculty of Medicine Siriraj Hospital, Mahidol University, Bangkok, Thailand
[3] Division of Bioinformatics and Data Management for Research, Department of Research and Development, Faculty of Medicine Siriraj Hospital, Mahidol University, Bangkok, Thailand
[4] Medical Biotechnology Research Laboratory, The National Center for Genetic Engineering and Biotechnology, National Science and Technology Development Agency, Pathumthani, Thailand
[5] Division of Dengue Hemorrhagic Fever Research, Department of Research and Development, Faculty of Medicine Siriraj Hospital, Mahidol University, Bangkok, Thailand

Corresponding authors
Prapat Suriyaphol,
prapat.sur@mahidol.ac.th
Dumrong Mairiang,
dumrong.mai@biotec.or.th

## ABSTRACT

Development of high-throughput technologies, such as Next-generation sequencing, allows thousands of experiments to be performed simultaneously while reducing resource requirement. Consequently, a massive amount of experiment data is now rapidly generated. Nevertheless, the data are not readily usable or meaningful until they are further analysed and interpreted. Due to the size of the data, a high performance computer (HPC) is required for the analysis and interpretation. However, the HPC is expensive and difficult to access. Other means were developed to allow researchers to acquire the power of HPC without a need to purchase and maintain one such as cloud computing services and grid computing system. In this study, we implemented grid computing in a computer training center environment using Berkeley Open Infrastructure for Network Computing (BOINC) as a job distributor and data manager combining all desktop computers to virtualize the HPC. Fifty desktop computers were used for setting up a grid system during the off-hours. In order to test the performance of the grid system, we adapted the Basic Local Alignment Search Tools (BLAST) to the BOINC system. Sequencing results from Illumina platform were aligned to the human genome database by BLAST on the grid system. The result and processing time were compared to those from a single desktop computer and HPC. The estimated durations of BLAST analysis for 4 million sequence reads on a desktop PC, HPC and the grid system were 568, 24 and 5 days, respectively. Thus, the grid implementation of BLAST by BOINC is an efficient alternative to the HPC for sequence alignment. The grid implementation by BOINC also helped tap unused computing resources during the off-hours and could be easily modified for other available bioinformatics software.

## INTRODUCTION

Massive data are now affordably, easily and frequently generated by genomic and proteomic assays such as massively parallel sequencing and high-throughput mass spectrometry. Up to 1 trillion bases can be sequenced in one 6-day run by Illumina HiSeq 2500 (*Rhoads & Au, 2015*) while mass spectrometry can now completely analyse a proteome and quantify protein concentrations in an entire organism (*Ahrne et al., 2015*). Breakthroughs in genomic and proteomic data generation lead to development and emergence of several disciplines. In precision medicine, clinicians can diagnose and tailor a treatment for a disease based on the patient profile derived from "omics" data (*Chen & Snyder, 2012*). Furthermore, metagenomics, a study of genetic materials in samples directly collected from particular environments, is now greatly advanced by high-throughput assays, and now becomes applicable to forensic sciences (*Fierer et al., 2010*) and pathogen discovery (*Chiu, 2013*). However, genomic and proteomic data are not readily usable or meaningful without proper analysis and interpretation which become the bottleneck of genomic and proteomic studies due to tremendous computational resource requirement (*Scholz, Lo & Chain, 2012*; *Berger, Peng & Singh, 2013*; *Neuhauser et al., 2013*).

To overcome the bottleneck of data analysis, high performance computing (HPC) is now commonly used in large-scale bioinformatics tasks including sequence alignment (*Orobitg et al., 2015*), simulation (*Zhang, Wong & Lightstone, 2014*) and machine learning (*D'Angelo & Rampone, 2014*). The physical architecture of HPC consists of numerous processing units, large shared memory and huge data storage cooperatively functioning to obtain high performance usually measured as floating-point operations per second (FLOPS) (*Subramaniam & Feng, 2012*). However, physical HPC is costly and requires extensive maintenance. Cloud computing services, such as Amazon EC2, is now an alternative to purchase a physical HPC for scientific computing (*Juve et al., 2009*). Several bioinformatics applications and frameworks are now designed to utilize cloud computing and/or grid computing such as CloVR (*Angiuoli et al., 2011a*), Galaxy (*Blankenberg et al., 2010*), Tavaxy (*Abouelhoda, Issa & Ghanem, 2012*) and CloudBurst (*Schatz, 2009*). Bioinformatics analyses usually involve repetitive computing intensive tasks that can be split into several smaller and less computing intensive tasks (*Carvalho et al., 2005*). Thus, massive parallelization on HPC or cloud computing was usually employed for large-scale bioinformatics analyses. MapReduce framework, such as Apache Hadoop (*White, 2012*), is usually employed for massive parallelization in which input data are split and mapped to worker nodes while output data from worker nodes were merged or reduced at the head node (*Dean & Ghemawat, 2008*). Recently, Apache Spark has become another framework for parallelization and cluster computing (*Zaharia et al., 2010*). Nevertheless, both physical HPC and cloud computing are still too costly for some research groups and require personnel with advanced computing skills to manage the systems.

There are also large public grid systems such as Open Science Grid (*Pordes et al., 2007*) that were used for bioinformatics data analysis including BLAST analysis (*Hayashi et al., 2014*). However, large inter-institutional grid systems may not always be suitable under certain circumstances. For example, limited bandwidth and firewalls can hinder the

 

data transfer to and from remotely located grid systems. In addition, clinical data must be carefully protected so transferring them to the public grid system may risk privacy violation. Also, if a grid system requires a dedicated server with routine maintenance, it will not be affordable for a small research group. A local grid system, which could be easily assembled, disassembled and then reassembled, will benefit a research group that could intermittently access abundant, but not individually powerful, computing resources such as a computer training center.

Berkeley Open Infrastructure for Network Computing (BOINC) is a middleware that helps manage volunteer and grid computing (*Anderson, 2004*). One of the best known BOINC-based projects is SETI@home whose purpose is to search for signs of extra terrestrial intelligence from radio telescope data (*Anderson et al., 2002*). SETI@home project recruits home computers to help analyse small chunks of data during the idle time. Data are sent and received through the internet creating a large distributed computing system. BOINC also supports an implementation of grid computing on local desktop computers (*Balaton et al., 2007*). Academic and research institutes usually own several desktop computers, which are idle during the off-hours. These unused computer resources can be tapped by implementing grid computing with BOINC. The BOINC-based grid computing could inexpensively provide adequate computing power required by many bioinformatics analyses.

Sequence alignment is one of the basic analyses for genomic and proteomic data and arranges DNA, RNA or protein sequences against one another or sequence databases to detect similarities or differences in order to infer functional, structural, or evolutionary relationships (*Baxevanis & Ouellette, 2001*). Basic Local Alignment Search Tools (BLAST) (*Altschul et al., 1990*) is a program widely used for sequence alignment. However, BLAST is not optimized for analysing massive data generated by high-throughput assays, and using BLAST for high-throughput data may lead to impractical runtimes (*Li, Ruan & Durbin, 2008*; *Borozan, Watt & Ferretti, 2013*). Thus, other sequence alignment programs have been developed such as BWA (*Li & Durbin, 2009*) and Bowtie (*Langmead et al., 2009*) to handle short sequences generated by high-throughput assays. As these methods are mostly specialized to only handle short sequences, several methods have been applied to improve the performance of BLAST in order to handle large data such as using parallel processing (*Darling, Carey & Feng, 2003*), grid computing (*Carvalho et al., 2005*) and cloud computing (*Angiuoli et al., 2011b*).

This study did not, however, aim to develop a novel method to increase the speed of sequence alignment. We aimed to use BLAST as a model for grid implementation because it is one of the most commonly used bioinformatics tools (*Carvalho et al., 2005*) with good documentation. Furthermore, BLAST can be performed in parallel without the need to communicate among worker nodes during processing (*Mathog, 2003*). There was a study by *Pellicer et al. (2008)*, that had applied BOINC to BLAST to increase the alignment speed. Our study aimed to confirm that the grid implementation with BOINC proposed by Pellicer et al., functioned well with actual next-generation sequencing data. In addition, we aimed to document instructions on how to set up BOINC grid system. The BLAST model for grid computing using BOINC is the first step to evaluate and design a simple

grid implementation, which will help research groups with limited computing resources tap into idle computers in their organizations for large-scale bioinformatics analyses.

## MATERIALS AND METHODS

### Sequences and databases

To test sequence alignment with BLAST, human DNA sequencing data (genome ID: NA12878) was downloaded from the Genome in the Bottle Consortium (ftp://ftp-trace.ncbi.nlm.nih.gov/giab/ftp/data/NA12878/NIST_NA12878_HG001_HiSeq_300x) (*Zook et al., 2014*). NA12878 data were from Illumina HiSeq2500 sequencing platform with about $300\times$ total coverage of $150 \times 150$ bp. The data contained 4 million sequence reads in fastq format. However, fastq format was not compatible with the BLAST program so the data were converted to fasta format using seqIO command of Biopython (version 1.65) (*Cock et al., 2009*) (https://github.com/dummai/BoincBlastTest/blob/master/PyScripts/convertFastqToFaQual.py). Since it would take an impractical amount of time to perform BLAST analysis on all 4 million sequence reads, the data were split into smaller chunks including read 1 only, read 1–10, read 1–100, read 1–1,000, read 1–10,000 and read 1–100,000. For task distribution by BOINC, the whole data were split into 40,000 smaller files each containing 100 reads with Chunk 1 containing read 1 to read 100, Chunk 2 containing read 101 to read 200 and so forth using a Python script (https://github.com/biopython/biopython.github.io/blob/master/wiki/Split_large_file.md). NA12878 data were aligned to the human genome, Genome Reference Consortium Human Reference 38 (hg38), downloaded from UCSC Genome Bioinformatics (http://hgdownload.cse.ucsc.edu/goldenPath/hg38/bigZips/) (*Rosenbloom et al., 2015*). The human genome (hg38) were formatted with the *makeblastdb* command of BLAST program.

### BLAST analysis

NCBI BLAST+ (version 2.2.30) for Microsoft® Windows 32-bit and for Linux/GNU 32-bit were downloaded from the National Center for Biotechnology Information repository (ftp://ftp.ncbi.nlm.nih.gov/blast/executables/blast+/2.2.30/) (*Camacho et al., 2009*). *blastn* command of NCBI BLAST+ was used to align NA12878 sequences against hg38. The output format was set to tabular with comment lines (-outfmt 7) and filtering with dust was turned off (-dust no). Finally, any hit that was enveloped by at least 20 higher-scoring hits was deleted (-culling_limit 20). The same *blastn* command line with the aforementioned settings was used for sequence alignment on all platforms. Read 1 only, read 1–10, read 1–100 and read 1–1,000 were used for BLAST analysis on a desktop computer. Read 1–10,000 and read 1–100,000 were additionally used for BLAST analysis on a HPC. All sequence alignments, except for read 1–100,000 set on HPC, were conducted in triplicate to derive an average processing time.

### Computers for BLAST analysis

The specification of the computer used for BLAST analysis was Intel® Core™ i7-4500U CPU @2.40 GHz with 8.00 GB of RAM. To simulate resources used per one workunit

on BOINC grid system, hardware virtualization was required to allocate one processing unit and 1.00 GB of RAM. The virtualization was done by using Oracle® VM VirtualBox (version 5.0.8). The operating system for virtualization was Microsoft® Windows 7 (32-bit).

### HPC for BLAST analysis

The specification of the HPC used for BLAST analysis was Cisco® UCS Blade Server B200M2 $\times$ 2 Units with CPU 2 $\times$ 6 cores (2.4 GHz, 12 cores in total) and 96 GB of RAM. The HPC could use up to 24 processing threads simultaneously. The operating system of the HPC was Ubuntu Server Linux/GNU (version 14.04 TLS). BLAST analysis was restricted to run on 12 processing threads by adding '-num_threads 12' to the *blastn* command line.

### Computer training center for grid implementation

A computer training room with 49 desktop computers was used for grid system implementation. The room was accessible from 5 pm to 8 am, after its regular operating hours. The specification of each computer was Intel® Core™ i5 CPU 660 @3.47 GHz with 4.00 GB of RAM. The operating system of all computers was Microsoft® Windows 7 (32-bit). Every computer was connected to local area network. One computer was assigned as a host machine for the BOINC project server. This computer had Oracle® VM VirtualBox (version 5.0.8) installed to virtualize hardware for the project server. The virtual machine image of the BOINC project server (version April 12, 2014 on Debian Linux/GNU version 7) was downloaded from BOINC webpage (https://boinc.berkeley.edu/dl/debian-7-boinc-server-140412.7z). One processing unit and 512 MB of RAM were allocated for the project server. Other computers were assigned as client machines with BOINCManager (version 7.4.42; https://boinc.berkeley.edu/dl/boinc_7.4.42_windows_intelx86.exe) installed.

### Grid implementation by BOINC

Documentation for grid implementation by BOINC are available at https://github.com/dummai/BoincBlastTest. BLAST program must be adapted to BOINC system for distributed computing. Thus, BOINC wrapper program (version 26014 for Microsoft® Windows 32-bit; https://boinc.berkeley.edu/dl/wrapper_26014_windows_intelx86.zip) along with an XML script (https://github.com/dummai/BoincBlastTest/tree/master/xml) were applied to the batch file containing a command line of *blastn* program and time recording commands (https://github.com/dummai/BoincBlastTest/blob/master/bat/blastn_windows_intelx86_0.bat). An XML script was used as a template for how to send clients input data including the hg38 database and each chunk of sequencing reads (https://github.com/dummai/BoincBlastTest/blob/master/xml/blastn_wu). Another XML script was used as a template for how to return BLAST analysis result back from the clients (https://github.com/dummai/BoincBlastTest/blob/master/xml/blastn_re). Python scripts were used to create bash shell scripts for staging input files (https://github.com/dummai/BoincBlastTest/blob/master/PyScripts/stage_file.py) and creating workunits (WUs; https://github.com/dummai/BoincBlastTest/blob/master/PyScripts/create_wu.py). One WU is a job package containing input data, application and instruction to be sent and processed at the client machines. Since this is a grid computing system, an option to

create redundant WUs for results verification required in BOINC volunteer system was turned off. Due to limited time allocation for accessing the training center, only 5,000 files containing 500,000 sequence reads were tested during one overnight run. The overall processing time was calculated from the difference between the time the first WU was sent and the time the last result received. In addition, the time points at which each WU was sent out, started processing, finished processing and retrieved back were recorded. Since a lagging process on a single machine could delay the overall processing time, a time limit of one hour was set to terminate a process taking longer than one hour and resend to other machine. To ensure that the grid implementation did not affect the results of BLAST analysis, the sequence alignment results of read 1–1,000 from the desktop computer and BOINC grid were compared.

## RESULTS AND DISCUSSION

### Sequence alignment without grid implementation

Sequence alignment with BLAST program could be performed on a web-based application at NCBI webpage (http://blast.ncbi.nlm.nih.gov/Blast.cgi). However, this is not a suitable option for analysing a large number of query sequences due to its dependence on network connection for query submission and result download. In addition, NCBI does not provide computer resources for all large-scale analyses submitted through the web-based application. Thus, standalone BLAST is provided by NCBI for running sequence alignment locally on a computer or HPC. In this study, standalone BLAST was chosen to benchmark analysis time of a desktop computer and HPC against that of BOINC grid system.

From a preliminary run of BOINC grid system, we found that one client machine could process up to four WUs simultaneously (data not shown). This was later confirmed when full grid implementation by BOINC was tested (Fig. 5). To benchmark the processing time of a single computer against BOINC grid system, we restricted computing resources for running standalone BLAST on the computer by hardware virtualization to the same resources used for processing one WU in BOINC grid system, which included one central processing unit and 1.00 GB of RAM. Additionally, the computing resources restriction helped estimate the time required for processing one WU with various number of sequence reads on BOINC grid system. Thus, an optimal number of sequence reads per WU could be selected. One, ten, 100 and 1,000 sequence reads were aligned against the human genome (Fig. 1). The average processing time for sequence alignment of 1,000 reads was 204.17 min which was too long for processing one WU. An error at the end of processing a WU with 1,000 reads would result in a loss of three and a half hours of processing time. Thus, WU's with 100 reads was used for analysis on BOINC grid, which would take approximately 20 min per WU. Simple linear regression analysis was used to predicted the analysis time for all 4 million reads. Based on an estimated 12.28 s to analyse a single read, it would take approximately 568 days to complete the sequence alignment with a linear approach. However, multiprocessing and multithreading capabilities of the current personal computer would significantly reduce the overall processing time depending on its central processing unit and shared memory.

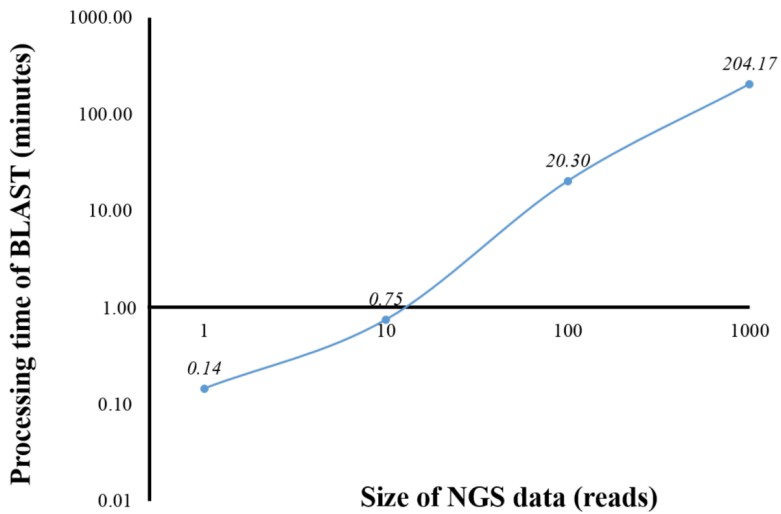

**Figure 1 Processing time of BLAST sequence alignment by number of reads analyzed on a single personal computer.** One, 10, 100 and 1,000 reads were analyzed in triplicate. The times shown were average processing times of triplicate runs.

Currently, HPC is used in several bioinformatics analyses (*D'Angelo & Rampone, 2014*; *Orobitg et al., 2015*; *Zhang, Wong & Lightstone, 2014*). HPC can manage large datasets and handle intensive computation while significantly reduce the processing time. However, HPC is expensive and requires extensive maintenance. In this study, HPC was used for benchmarking the time used for sequence alignment against that of the alignment on BOINC grid. Twelve threads of HPC were simultaneously used for BLAST program. From the initial dataset of 4 million reads, one, ten, 100, 1,000, 10,000 and 100,000 sequence reads were aligned against the human genome (Fig. 2). The sequence alignment of 100,000 reads took about 880 min. Simple linear regression analysis was used to predict the relationship between number of reads and total analysis time on the HPC. An estimated 0.53 s per read meant it would take approximately 24 days to complete the sequence alignment for all 4 million reads. The HPC used in the study was a small model, however, a larger HPC might be able to complete the same sequence alignment within a few days.

## Sequence alignment with grid implementation by BOINC

BOINC helps manage distribution of a large number of tasks to client machines on volunteer or grid system. In addition, BOINC has a system to validate the results returned by the clients. However, the BOINC project administrator is required to prepare input data for task distribution as well as plan methods to consolidate result files into a single final result. In this study, 4 million sequence reads were split into 40,000 files with each file containing 100 sequence reads. The number of files was decided to minimize the number of files while keeping total runtime at a reasonable level (Fig. 1). The human genome database was also sent to client machines along with a sequence file. However, a "sticky" option of BOINC ensured that the database was transferred to the same client only once and remained in the machine for other rounds of sequence alignment.

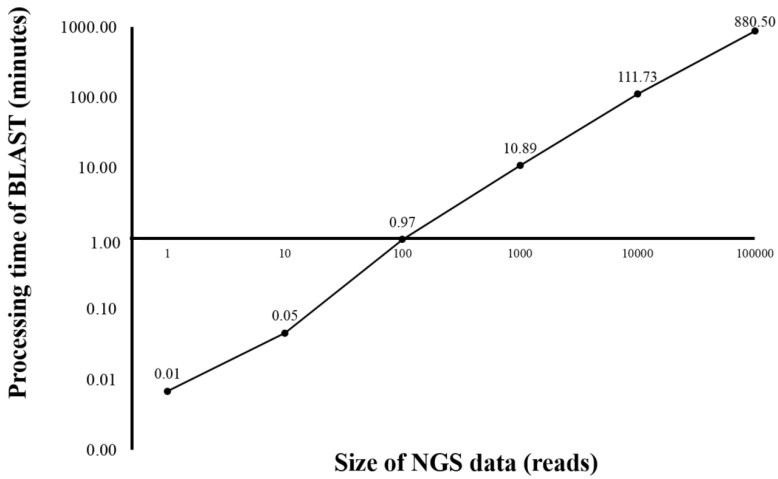

**Figure 2  Processing time of BLAST sequence alignment by number of reads analyzed on HPC.** One, 10, 100, 1,000 and 10,000 reads were analyzed in triplicate. The times shown were average processing times of triplicate runs. Only one run for BLAST analysis was performed for 100,000 reads due to an extensive processing time.

In this study, BOINC grid system was tested in one overnight session. One session was expected to be sufficient for sequence alignment of 500,000 reads. Consequently, Chunk 1 to Chunk 5,000 of data, which contained read 1 to read 500,000 were used during this session. The time between when the first file was sent to the client and the time the last result file was received by the project server was 846 min or about 14 h. The cumulative elapsed time from each WU was 90,400 min meaning the grid system reduced the real processing time by a factor of 107. The average processing time of the WUs was 18 min with a range of 5–48 min (Fig. 3). The client machines showed a bimodal distribution in running time, with one group having an average processing time per WU at about 16–18 min while a minority group had a longer processing time per WU at more than 23 min (Fig. 4A). To measure whether data transfer speed via the local area network significantly affected the processing time at each client machines, the runtime of each blast analysis were also recorded at the client machines. The difference between the processing time recorded at the server and that at the client machine was the time used for data transfer. The average data transfer time was 41 s with a range of 16–219 s (Fig. 4B). The delay caused by data transfer speed was small compared to the actual times used for BLAST analysis. Thus, data transfer speed did not affect the processing time observed in the client machines. It was impossible to directly measure the times used for sending or receiving data since the clocks of the project server and client machines were not synchronized. During the overnight session, an average number of WUs processed by each client was 104 with a range of 92–140 WUs (Fig. 4C) resulting in an average of 10,400 processed reads. A minority of client machines had processed 130 WUs or more (Fig. 4C). Interestingly, the minority of client machines with longer average processing times and the client machines with more WUs processed were exactly the same machines. When the number of WUs being processed at a particular time was tracked, we found that the slow group simultaneously

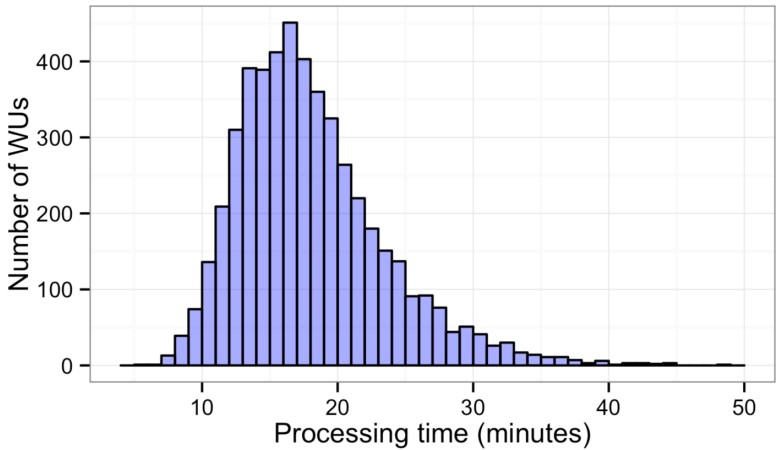

**Figure 3** **Histogram of sequence alignment times for each WU.** Five thousand WUs were distributed through a BOINC grid system with 50 client machines in one 14-hour-long session. Each WU contained 100 sequence reads of the total 500,000 reads.

processed 4 WUs most of the time while the fast groups only simultaneously processed 2 WUs most of the time. The machine with the longest average processing time and the one with the shortest processing time were selected for tracking the number of WUs being processed in Fig. 5. We suspected that the configuration of hardware (other than CPU and RAM) and/or software (background application) of these two groups were different. BOINC grid system is designed to cope with clients with heterogeneous configurations and specifications so the difference in the ability to process WUs in parallel would not cause a major problem.

The sequence alignment of read 1–1,000 from BOINC grid (https://github.com/dummai/BoincBlastTest/blob/master/raw/BOINC_1000reads_BLASTresults.txt) were compared to the results from the single computer (https://github.com/dummai/BoincBlastTest/blob/master/raw/singlePC_1000reads_BLASTresuls.txt). The results were identical. Thus, the BOINC grid system helped increase the speed of sequence alignment greatly and did not affect the final results. Using the same BOINC grid setup, it would take 8 overnight sessions or one 5-day-long session to finish the sequence alignment of 4 million reads. The BOINC grid is highly scalable and when more clients join the grid the overall processing speed of the grid would proportionally increase.

There are other middleware or systems for grid implementation including Condor (*Epema et al., 1996*), Univa Grid Engine (http://www.univa.com/products/) and PBS/Torque (http://www.adaptivecomputing.com/products/open-source/torque/). We did not select Univa Grid Engine because it is proprietary. PBS/Torque system is not compatible with a computer training center with machines running on Microsoft Windows. Condor, now known as HTCondor, is a comparable alternative to BOINC for grid implementation in a computer training center environment. Condor has an advantage over BOINC as the modification of an application to be distributed is not required if the application has binary compatibility (*Søttrup & Pedersen, 2005*). However, Condor is not suitable with volunteer computing that requires validation strategy whereas BOINC utilizes
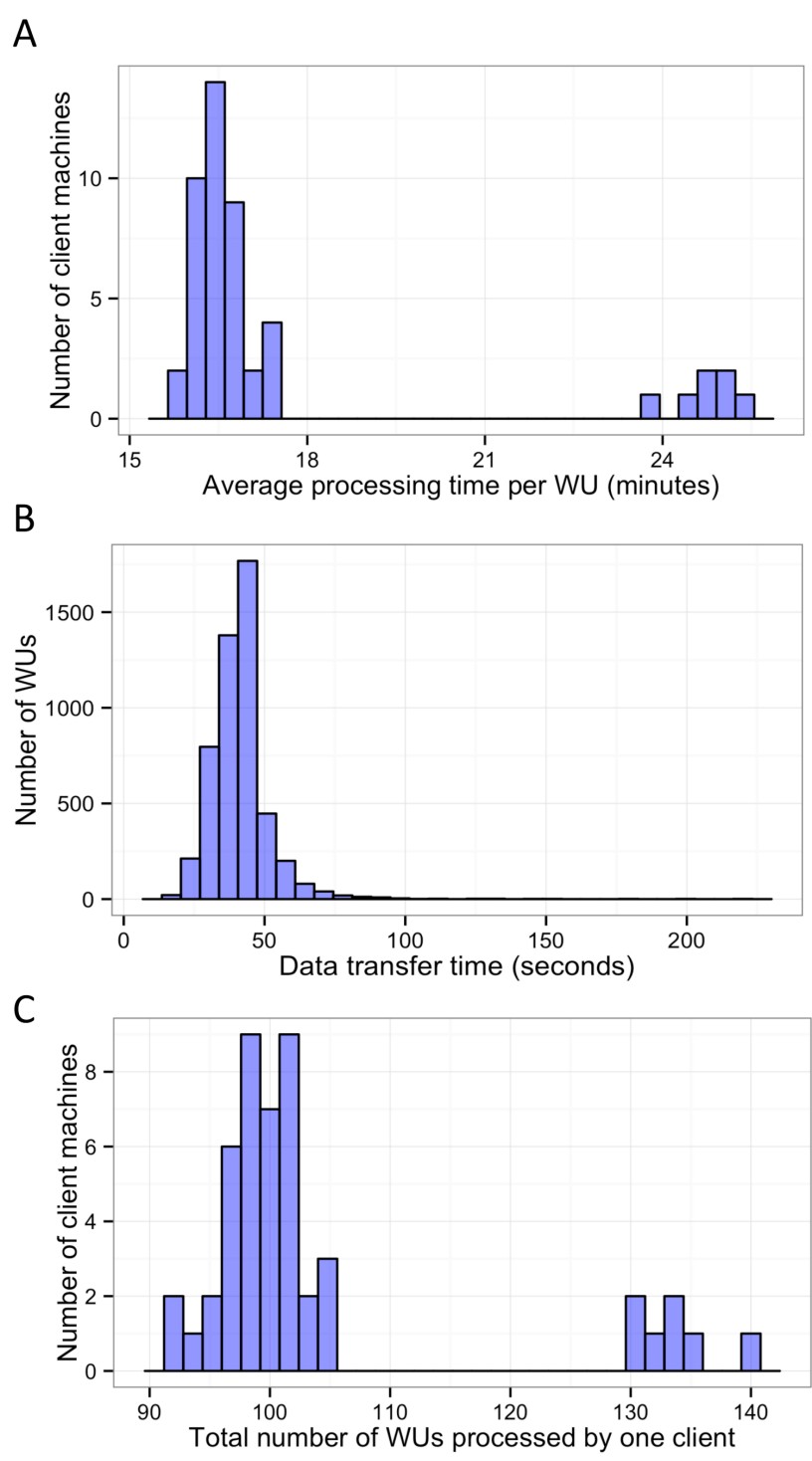

**Figure 4  Histograms of the performance of client machines during one 14-hour-long session of BLAST analysis through BOINC grid system.** Average processing time per WU of each client machine was measured (A). Time used for data transfer of each WU was recorded (B). Finally, the total number of WUs each client machine handled during the sessions was recorded (C).

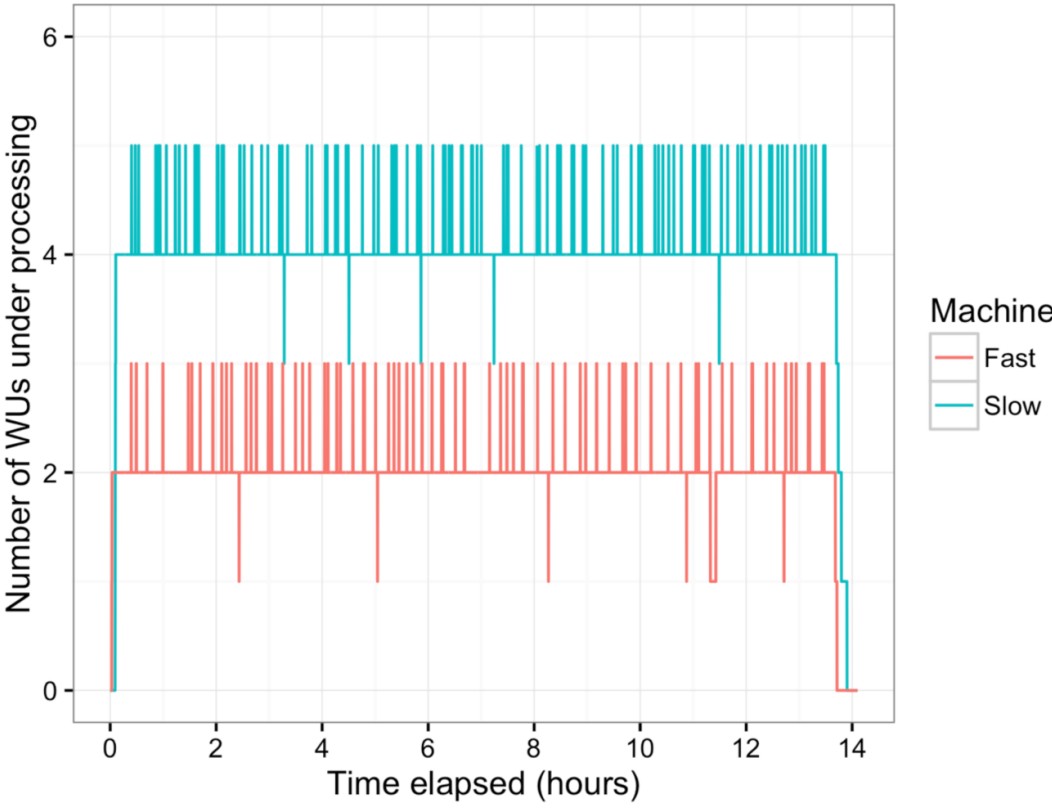

**Figure 5  Number of WUs under processing at a particular time.** Two machines with the longest (Slow, blue line) and the shortest (Fast, red line) average processing time per WU were tracked for the number of WUs under processing in parallel during the overnight session. Small peaks happened as the transfer of completed WUs and new WUs were overlapping.

homogenous redundancy to do so (*Søttrup & Pedersen, 2005*). In this study, BOINC was selected because there were more projects similar to this study available as reference and instructions of the initial installation and configuration were available.

BOINC grid system still has some limitations. In metagenomics analyses, sequencing data might be aligned against the nucleotide collection database (nr database). The nr nucleotide database is large and it would be difficult to distribute the database to each client. A strategy for splitting nr database and merging alignment results would be necessary. BOINC grid does not allow clients to directly contact one another. Thus, some applications that require client communication would not be suitable for grid implementation with BOINC. On the other hand, the BOINC grid system allows clients with various operating systems as long as they are compatible to the application used in the grid system. The wrapper program for BOINC provides convenient methods to adapt applications for grid implementation.

We created a series of documentation for setting up of BOINC grid system to help other research groups with limited computing resources tap unused computer in their institutes (https://github.com/dummai/BoincBlastTest).

## CONCLUSIONS

Many academic and research institutes have under-utilized computing resources in the form of computer training centers. In this study, we showed a way to potentially tap these unused resources by grid implementation with BOINC. We tested the grid system using BLAST analysis as a model. The results showed that the grid system greatly increase the speed of BLAST analysis without affecting the quality of the results. Thus, the grid implementation with BOINC would be an economically alternative to HPC for any research groups with limited resources.

## ACKNOWLEDGEMENTS

We would like to thank Associate Professor Thawornchai Limjindaporn for his valuable advice on this project and Dr. Harald Grove for very helpful English editing.

### Funding

The authors received no funding for this work.

### Competing Interests

The authors declare there are no competing interests.

### Author Contributions

- Watthanai Pinthong performed the experiments, analyzed the data, contributed reagents/materials/analysis tools, wrote the paper, prepared figures and/or tables, reviewed drafts of the paper.
- Panya Muangruen performed the experiments, contributed reagents/materials/analysis tools, reviewed drafts of the paper.
- Prapat Suriyaphol conceived and designed the experiments, contributed reagents/materials/analysis tools, reviewed drafts of the paper.
- Dumrong Mairiang conceived and designed the experiments, performed the experiments, analyzed the data, contributed reagents/materials/analysis tools, wrote the paper, prepared figures and/or tables, reviewed drafts of the paper.

### Data Availability

Codes have been deposited in GitHub: https://github.com/dummai/BoincBlastTest.

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
