# Peer review of "A simple grid implementation with Berkeley Open Infrastructure for Network Computing using BLAST as a model"

_PeerJ, doi:10.7717/peerj.2248_

## Round 0.1 · original submission · Major Revisions

· Academic Editor

Major Revisions

Please take a close look at the comments by reviewer #2. They correctly point out some major weaknesses that need to be addressed prior to publication.

Reviewer 1 ·

Basic reporting

The article is well written but there are a number of grammatical errors that need correction and these are included below:

Line 49: Consider revising “genomic and proteomic data generations lead to development and emerging of several disciplines” to “genomic and proteomic data generation lead to development and emergence of several disciplines”

Line 59: Consider revising “(HPC) are” to “(HPC) is used” as HPC is singular.

Literature referenced: The existing literature has been researched and cited

Line 81: Consider revising “SETI@home of which purpose is” to “SETI@home whose purpose is”

Line 104: Consider revising “good documentations” with “good documentation”

Line 142: On line 142 there are two clock speeds specified for the desktop computer. Please correct.

Line 148: While the hardware system for HPC is identified the clock speed for the UCS system is not specified. Please add clock speed and core count.

Line 165: Consider replacing “distribution computing” with “distributed computing”

Line 194: Consider revising the sentence to read “To benchmark the processing time of a single computer”
Line 214: Consider revising the sentence to read “Twelve threads of HPC”

Line 220: Consider revising to read “The HPC used in the study was a small model, however, a larger HPC might be able to complete the same sequence alignment within a few days.”

Line 236: Consider revising sentence to read “With 50 machines if each machine processed one WU at a time”‚ i.e., replace “process” with “processed”

Line 240: Consider revising the sentence to read “The time between when the first file was sent to the client”

Line 243: Consider revising sentence to “When we measure the performance of client machine”

Line 244: Consider revising to “minority of client machines had longer processing times”

Line 268: Consider revising sentence to “would be difficult to completely distribute the database”

Line 274: Consider revising to read “convenient methods to adapt applications for grid implementation”

Line 281: Consider revising to “have unused computing resources in the form of”

Line 261: Consider dropping “Nevertheless” and revise to read “The BOINC grid is highly scalable and when more clients join the grid the overall processing speed of the grid would proportionally increase.”

I have not seen a link to the software or the raw data from the experiment.

Pellicer study: The authors introduce the Pelicer study later in the paper. Given that this study is a slight modification on that study it should be introduced earlier in the paper.

Page 12
Limitations: Consider moving the limitations to a separate paragraph

Add commas to numbers: To improve readability please consider adding comma separators to numbers. For instance 100000 would be 100,000.

Page 19
Figures and legends are mixed: The title of Figure 1 has Figure 2 in the legend and vice versa

Experimental design

The experimental design is well thought out and the experiment conducted in a robust manner. There are few things where the experiment and manuscript could be improved and include:

Page 6
Rationale for why BOINC? Could the authors comment on the benefits of BOINC over other grid management systems such as Condor, Univa Grid Engine, or PBS/Torque.

Page 8:
Data preparation
Not very clear about how the data were chunked? It appears that for local desktop and HPC benchmarks a subset of the data was used while for grid benchmarking the entire dataset was used. May consider clarifying that.

Calculating Grid times: Use of processing time from first to last job, or total elapsed time, is a good measure but can be affected by a single lagging process. Would a better measure be the cumulative elapsed time for each WU?

Clock speeds: How did the investigators normalize for differing CPU clock speeds and CPU models? These two can significantly affect performance.

Benchmarking of single computer: Reading the results section it appears that the single computer benchmarks were generated on the BOINC system by restricting the number of processors. But in the “methods and materials section” a single computer with different specifications was mentioned. Could the authors clarify which it was?

Page 21
Average processing time: Was the bi-modal distribution a reflection of sequence composition of hardware. One would have expected a more even distribution as opposed to a bimodal distribution

Validity of the findings

The data is robust and the experiments repeated a sufficient number of times to generalize the results. However, one thing that needs clarification is that the authors use Linux for HPC and Windows on the grid. Could they comment on the run times of blast on the same data on similar systems to ensure that the OS is not affecting the benchmarking?

Additional comments

Overall this is an important study that will benefit small labs that do not have access to large computational infrastructure.

·

Basic reporting

The authors present their study for using BLAST for diverse sequence alignments exploiting BOINC. While this is a worthwhile approach, it is not really clear what the authors have added at this state of their study except for some novel performance results upon the findings of Pellicer et al., whom they also cite. They mention solutions like Galaxy but only in regard to Cloud solutions, not for grid. Galaxy can be also connected to grid infrastructures. There are also some missing references to solutions like Tavaxy, which has been tested for similar use cases and BLAST (see http://www.ncbi.nlm.nih.gov/pmc/articles/PMC3583125/) or BLAST optimizations and performance studies for the Open Science Grid (http://pos.sissa.it/archive/conferences/210/025/ISGC2014_025.pdf)
The manuscript lacks a performance comparison to other grid infrastructures like national grid infrastructures at all. The authors describe their intention to improve data mining and simulation methods in the future and I recommend that they continue this work to a mature state, which can be published.

Experimental design

The authors describe clearly the hardware and data sets used.

Validity of the findings

While the performance measures for the runs look good, the authors may want to optimize more regarding the underlying database and improve their approach upon the findings, which have been already published by Pellicer et al.

Additional comments

The manuscript needs more proofreading and contains a couple typos like chuck instead of chunk or Pelicer instead of Pellicer.

---

## Round 0.2 · accepted · Accept

· Academic Editor

Accept

The revisions to the article are sufficient.

·

Basic reporting

The comments have been addressed or answered. I recommend the article for publication now.

Experimental design

No further comments

Validity of the findings

No further comments